# Extremely Low Leakage Expression Systems Using Dual Transcriptional-Translational Control for Toxic Protein Production

**DOI:** 10.3390/ijms21030705

**Published:** 2020-01-21

**Authors:** Yusuke Kato

**Affiliations:** Division of Biotechnology, Institute of Agrobiological Sciences, National Agriculture and Food Research Organization (NARO), Oowashi 1-2, Tsukuba, Ibaraki 305-8634, Japan; kato@affrc.go.jp; Tel.: +81-29-838-6059

**Keywords:** tight control of gene expression, unnatural amino acid incorporation, riboswitch, ribozyme, STAR, antisense RNA, translational regulation, transcriptional regulation

## Abstract

Expression systems for highly toxic protein genes must be conditional and suppress leakage expression to almost zero because even faint leakage expression may kill host cells, inhibit host growth, and cause loss of plasmids containing the toxic protein genes. The most widely used conditional expression systems are controlled only at the transcriptional level, and complete suppression of leakage expression is challenging. Recent progress on translational control has enabled construction of dual transcriptional-translational control systems in which leakage expression is strongly suppressed. This review summarizes the principles, features, and practical examples of dual transcriptional-translational control systems in bacteria, and provides future perspectives on these systems.

## 1. Introduction

Heterologous gene expression is a fundamental technique for the production of recombinant proteins. A variety of bacteria are used as hosts for heterologous gene expression [1,2,3,4]. A problem with this technique is that about 50% of artificially expressed proteins are toxic to the host bacteria, even in self-gene overexpression [5,6]. This toxicity inhibits host bacterial growth and often kills host bacteria. In addition, loss of plasmids containing the toxic protein genes frequently occurs. To overcome this problem, conditional expression systems, in which toxic proteins are produced only in the presence of inducers, are widely used [7]. In the proliferative stage of host bacteria, the growth of bacteria is not affected because the production of toxic proteins is suppressed in the absence of inducers. After sufficient bacterial growth, the toxic proteins can be produced in the presence of inducers. However, the switchability of most “biological” conditional expression systems is not as perfectly tight as that of electronic switches. This means a small, but problematic, number of proteins are produced, even in the absence of inducers (leakage expression) [8]. Highly toxic proteins affect the growth rate and viability of host bacteria, even with faint leakage expression. Therefore, complete suppression of leakage expression is essential for production of active highly toxic proteins. However, this is a challenging problem.

Dual transcriptional-translational control, in which heterologous gene expression is regulated at the transcriptional and translational levels, has recently been developed to solve this problem in bacteria. In this review, the principles, features, and recent practical examples of dual transcriptional-translational control systems are described, together with a discussion of future perspectives on these systems.

## 2. Advantages of Dual Transcriptional-Translational Control

The most widely-used inducible expression systems are regulated at the transcriptional level only. In these systems, transcription is activated by an inducer, such as a specific chemical or physical stimulus. The operator/repressor system derived from the lactose operon in *Escherichia coli* (*lacO/I*), which is activated by lactose or its persistent analog, isopropyl *β*-_D_-1-thiogalactopyranoside (IPTG), is a classic example [9,10]. *LacO/I* continues to be used widely, but the level of leakage expression is greatly affected by promoter selection. Leakage expression of the native lactose promoter (*Plac*) and *lacO/I* pair is about 0.1% in an optimal setting, but reaches 2% in combination with *Ptac*, a stronger derivative of *Plac*, indicating insufficient tightness for expression of highly toxic genes [11]. The operator/repressor system derived from the arabinose operon (*araO/C*) in *Escherichia coli* is a widely used expression system that can tightly regulate transcription of a target gene [12]. Leakage expression of *araO/C* is <0.1%. However, a moderately strong promoter derived from the *araBAD* operon (*P_BAD_*) and an *araO/C*-regulated expression construct for the colicin E3 enzymatic domain (ColE3e), of which a few molecules can kill *E. coli* by cleaving 16S rRNA, cannot be maintained in a multicopy plasmid, indicating that the leakage level is too high [13,14,15]. Complete suppression of leakage transcription is still challenging despite the effort put into the development of tight transcriptional regulatory systems that permit functional expression of highly toxic proteins. Several conditional expression systems that work at the translational level have been reported, as described below, but complete suppression of leakage translation is also challenging in these systems.

Dual transcription-translation control provides a novel scaffold on which to construct an extremely tight expression system in which leakage expression is suppressed to the minimum. In conventional systems that regulate expression only at the transcriptional level, all leakage-expressed mRNAs are translated to toxic proteins; production of these mRNAs therefore must be zero to suppress the toxic protein production completely. In contrast, leakage mRNAs are faintly translated with additional translational suppression, and production of toxic proteins can be completely suppressed, even if transcription of the toxic protein gene is not perfectly suppressed. In other words, dual transcriptional-translational control may achieve extreme suppression of leakage expression, even if the transcriptional or translational regulation is leaky. In the following section, several dual transcriptional-translational control systems are described based on the order of the publication date.

## 3. Site-Specific Unnatural Amino Acid Incorporation

Incorporation of a site-specific unnatural amino acid (Uaa) into ribosomally synthesized proteins in vivo at a position encoded by an amber stop codon [16,17,18,19,20] was originally developed for structural analysis, labeling, chemical ligation, and functional modification of proteins through the replacement of canonical natural amino acids [16,17,18]. Site-specific Uaa incorporation can also be used to control translation of target mRNA (Figure 1A) [21,22,23,24]. Typically, amber stop codons are inserted near the translation initiation site of the coding region of target genes. Genes encoding the UAA-specific aminoacyl tRNA synthetase (UaaRS) and the cognate tRNA_CUA_ are also introduced into the host bacteria. Once the Uaa is provided in the culture medium, it is taken up into the intracellular space and then incorporated in proteins at the inserted amber stop codons, causing full-length translation of target mRNAs by amber stop codon read through (ON-state). In the absence of Uaa, translation is interrupted at the inserted amber stop codons, resulting in inhibition of functional target protein production (OFF-state).

Translational control using site-specific Uaa incorporation can regulate translation in an all-or-nothing manner, and also at any intermediate magnitude by adjustment of the Uaa concentration in the medium [22]. The EC_90_/EC_10_ ratio is 55 in translational control using a 3-iodo-_L_-tyrosine (IY) incorporation system derived from the tyrosyl-RS/tRNA pair of the archaeon *Methanocaldococus jannaschii*. Such intermediate-level expression is due to a uniform response of each individual cell, rather than changes in the population averages of induced and non-induced cells. 

The level of leakage translation varies among Uaa incorporation systems. For instance, leakage translation is only about 1% in the *N^ε^*-benzyloxycarbonyl-_L_-lysine (ZK) system, which was developed from pyrrolysyl-tRNA synthetase PylS and its cognate tRNA_CUA_ PylT of the archaeon *Methanosarcina mazei*, whereas that of the IY system is 6%–25% [21,22,23,25,26]. Various methods and settings can suppress the level of leakage translation. Lower expression of UaaRS and tRNA_CUA_ suppresses leakage translation, but yield also decreases when expression is too low [21]. Positive-feedback regulation of UaaRS and tRNA_CUA_ was recently developed to suppress leakage translation [23], through a mechanism in which sufficient amounts of UaaRS and tRNA_CUA_ are supplied in the ON-state, whereas expression of these genes is suppressed in the OFF-state. Such positive-feedback regulation enables lower leakage translation without a severe loss of yield. The IY system equipped with positive-feedback regulation achieved a gain (expression in the ON-state/leakage expression) of 1.4 × 10^3^ (synonymous with 0.07% leakage expression), which was 3 × 10^2^-fold higher than that of the parent system. This gain is comparable to that with *araO/C* [12]. Leakage translation is also suppressed through multiplexing of inserted amber stop codons, despite a loss of yield [21]. Some other methods have been proposed to suppress leakage translation, but experimental evidence is yet to be provided [24].

In 2014, the HYZEL (High-Yield and ZEro-Leakage) dual transcription-translation control expression system using Uaa incorporation was reported in *E. coli* (Figure 1A) [21]. In HYZEL, transcription of a toxic gene is controlled through a cascade under the T7 promoter (*P_T7_*) with *lacO/I* in the host bacterium BL21-AI, in which T7 RNA polymerase (T7RNP) is conditionally expressed under the control of *P_BAD_* and *araO/C*. The recombinant toxic protein is produced in the presence of _L_-arabinose, IPTG and Uaa (ON-state), whereas production is suppressed in the absence of these inducers and the presence of _D_-glucose, which causes catabolite suppression (OFF-state). HYZEL with translation of the toxic protein controlled by ZK incorporation tolerated an expression construct for ColE3e containing a single amber stop codon insertion, which suggests that leakage expression is almost zero (Kato Y, unpublished data, Figure 2A), while the yield of recombinant protein was not affected by insertion of 1 or 2 amber stop codons [21]. In addition, the DNA gyrase inhibitor, CcdB, a well-known toxic protein to *E. coli*, has been successfully produced using HYZEL (Kato Y, unpublished data, Figure 2B).

A drawback of HYZEL is that Uaa must be incorporated in the recombinant proteins, causing an unintended modification of the native amino acid sequence. This problem may be solved by using a Uaa designed to be incorporated in N-terminal tags or signal sequences that are removed after natural or artificial processing [21]. An alternative approach is use of a Uaa that can be converted into a natural amino acid by a chemical or biological process after incorporation into proteins [27,28].

## 4. Riboswitches

A riboswitch is a cis-regulatory element of mRNA that is usually located in the 5′ untranslated region (5′UTR) [29,30,31]. Binding of a specific ligand to the binding region of a riboswitch (aptamer domain) induces a conformational change that alters gene expression at the transcriptional or translational level. The Add-A translational ON riboswitch is an adenine-sensing riboswitch found in the bacterium *Vibrio vulnificus* [32]. In the absence of adenine, the ribosome-binding site (RBS) of mRNA is occluded by formation of a repressor stem in the riboswitch. In contrast, binding of adenine to the aptamer domain induces a conformational change in the riboswitch, resulting in initiation of translation by release of the RBS. A modified riboswitch, which is specifically controlled by the unnatural ligand pyrimido[4,5-d]pyrimidine-2,4-diamine (PPDA), has also been constructed and is referred to as the PPDA orthogonal riboswitch (PPDA-ORS) [33,34,35].

*RiboTite* is a tight conditional expression system that uses a combination of *P_T7_-lacO/I* controlled transcriptional regulation and PPDA-ORS-controlled translational regulation, as first reported in 2015 (Figure 1B) [36,37]. *RiboTite* is a cascade-regulatory system, similarly to HYZEL. In the tightest system among several variations of *RiboTite*, named *tT/tT*, the T7RNP gene is regulated by *lacO/I* at the transcriptional level and by PPDA-ORS at the translational level. A recombinant protein gene is transcribed by T7RNP and regulated by *lacO/I* and *PPDA-ORS*. *tT/tT* achieved low leakage expression (0.1%) and high gain (850-fold), suggesting that tightness was increased by >240-fold by addition of PPDA-ORS regulation, whereas the volumetric yield decreased by about 60%. In addition, the EC_90_/EC_10_ ratio increased from 10 in the parent system lacking PPDA-ORS regulation to 245 in *tT/tT*. Using *tT/tT,* the type II-like bacteriocin, epidermicin NI01 from *Staphylococcus epidermidis*, which has toxicity against *E. coli* in the cytoplasm, was produced at 2-fold higher than in the parent system [36,38]. Moreover, an expression construct of the toxic gene encoding levansucrase (*sacB*) was safely maintained in a modified *tT/tT* that omitted PPDA-ORS regulation of a recombinant protein gene, *tT/t*, in the presence of 5% sucrose, in which SucB had toxicity [36,39,40].

A dual transcriptional-translational control similar to *RiboTite* was independently developed in *Actinobacteria* and reported in 2016 (Figure 1C) [41]. This system has transcriptional regulation using *cymO/R*, an operator-repressor system that is derepressed by cumate (*p*-isopropylbenzoic acid), and translational regulation using a theophylline riboswitch, which regulates translation of mRNA by masking and releasing of the RBS, as well as an Add-A translational ON riboswitch [42,43]. The dual *cymO/R*-theophylline riboswitch control achieved a gain of 20- to 25-fold over that of a single control using a theophylline riboswitch only. No activity was detected in a leakage expression test using a *gusA* reporter, suggesting that this dual control is highly tight. Using dual *cymO/R*-theophylline riboswitch control, production of the macrodiolide antibiotic pamamycin, which is extremely toxic to streptomycetes, was successful in *Streptomyces albus* by tight control of *pamJ*, which encodes an essential enzyme for pamamycin biosynthesis [44,45]. Dual control using the operator-repressor system *rolR/O*, which is derived from the *Corynebacterium glutamicum* resorcinol catabolic operon, and the theophylline riboswitch has also been reported [41].

## 5. Ribozymes

Ribozymes are RNAs with catalytic activities [46,47], and some ribozymes are activated by binding of specific ligands. Theophylline ribozyme is a hammerhead ribozyme that recognizes and site-specifically digests a specific RNA sequence [48], and is activated in the presence of theophylline. Gene expression can be controlled at the translational level using this ribozyme. This control element, in which a complementary sequence to the RBS inhibits translation and the theophylline ribozyme, is inserted in the 5′UTR near the translation initiation site. Theophylline ribozyme removes this control element by self-digestion in the presence of theophylline, resulting in release of the RBS and translation initiation of mRNA. A dual control system using *cumO/R* and theophylline ribozyme was reported in 2016 (Figure 1D) [41]. No leakage expression was detected using a *gusA* expression construct regulated by this dual control system.

## 6. Antisense RNA

Antisense RNAs can regulate both transcription and translation *in trans* [49,50,51,52,53]. A dual transcriptional-translational control system regulated by a single antisense RNA was reported in 2017 (Figure 1E) [54]. This system was designed based on the regulatory element of the plasmid replication protein RepC in plasmid pT181 in the bacterium *Staphylococcus aureus* [55]. A transcriptional terminator is located in the 5′UTR near the translation initiation site of *repC*. Formation of the transcriptional terminator also inhibits translation of *repC* because the RBS is located in the terminator hairpin, similarly to the toehold switch, which suggests that terminator hairpin formation is a natural dual transcriptional-translational control element [51]. A specific controller antisense RNA, small transcription activating RNA (STAR), interrupts hairpin formation [50]. Expression of a recombinant protein gene with its coding sequence fused to the first 12 nucleotides of *repC* is controlled at the transcriptional and translational levels by a single regulator STAR. This dual control system achieved <0.1% leakage expression and a 923-fold gain.

A dual control system with antisense RNAs was reported in 2018, using combined STAR-regulated transcriptional control and antisense RNA-regulated translational control (Figure 1F) [56]. STAR and the antisense RNA are provided from transgenes regulated by a 3OC6-inducible *P_Lux_* promoter and a tetracycline-inducible *P_Tet_* promoter, respectively. In the ON-state, only expression of STAR is induced. In contrast, expression of antisense RNAs against STAR and the RBS and start codon is induced in the OFF-state, resulting in inhibition of transcription induction and translation initiation. With the optimal setting, 4% leakage expression was recorded for this system. 

## 7. Conclusions

Dual transcriptional-translational control is a powerful approach for achieving a minimal leakage system for the expression of toxic proteins. Some applications, such as ColE3e and CcdB in HYZEL, SucB and epidermicin in *RiboTite,* and pamamycin in the dual *cumO/R* and theophylline riboswitch control system, clearly support the potency of this approach. Dual control is also promising for other purposes in which low leakage is desired, such as switching of enzyme gene expression in metabolic engineering and biocomputing in synthetic biology [57,58]. For example, precise expression control of Cre recombinase, which was achieved using the dual *cumO/R* and theophyllin riboswitch control system, is useful both for genome editing and for non-volatile memory in biocomputing [41]. In addition, most dual control systems contain many regulatory factors, suggesting that Boolean multi-input logic gates can be constructed for integration of environmental signals in synthetic gene circuits [21,36,41,56]. However, such multi-regulator systems require many regulatory elements for control of a single target gene expression. This is a possible drawback because the number of well-characterized regulatory elements is limited, even in *E. coli.* Dual transcription-translation control using a single regulatory element may be an ideal solution to this problem (Figure 1E) [56].

Other forms of post-transcriptional regulation, such as mRNA degradation and antisense RNA against protein coding regions, may also be used to construct a dual or multi-control system [59,60]. Moreover, some methods have also been used to suppress leakage expression without improving the tightness of transcription or translation, such as inhibition of specific RNA polymerase activity in the OFF-state (e.g., lysozyme inhibition of T7RNP), phage delivery of a specific RNA polymerase after sufficient bacterial growth, and an adjustable plasmid copy number [5,61,62]. Construction of further improved tight control systems is likely to be achieved using multi-layered mechanisms working at different biological levels.

In conclusion, multi-layer control, such as the use of the dual transcriptional-translational control approaches described here, is a rational method to achieve tight regulation. This is the state-of the-art for building a reliable biological regulator out of unreliable components, indicating that the fuzziness of biological elements may be overcome to achieve a finely controlled biological system that behaves like an electronic device.

## Figures and Tables

**Figure 1 ijms-21-00705-f001:**
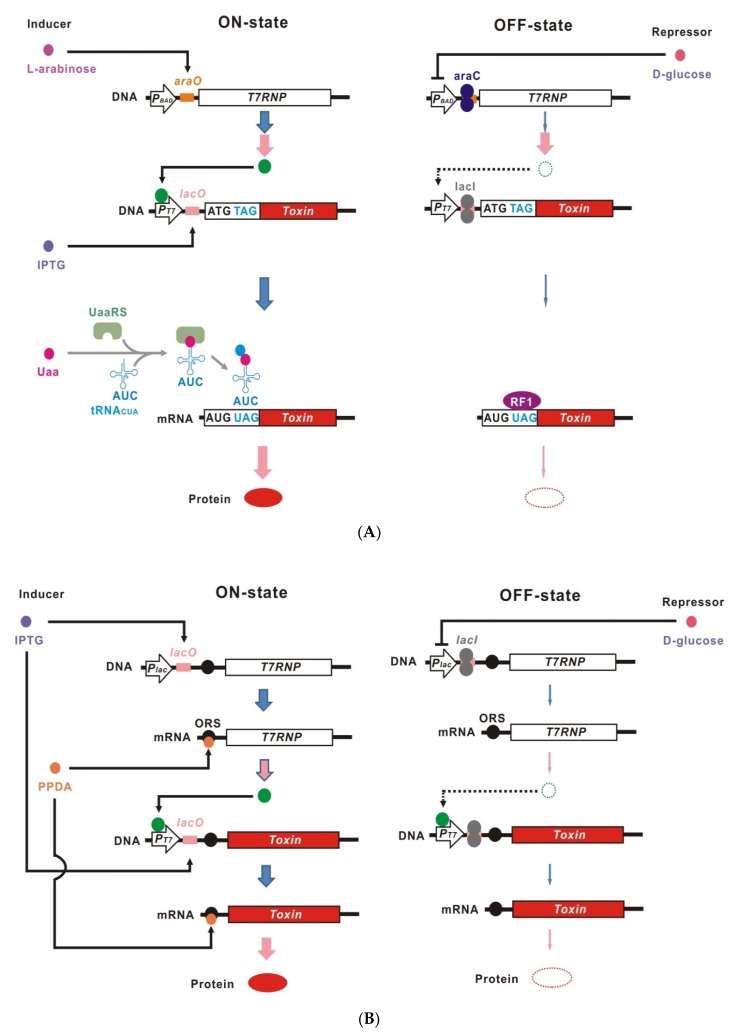
Architecture of dual transcriptional-translational control systems. Blue and pink arrows indicate transcription and translation, respectively. (**A**) HYZEL. RF1, peptide release factor 1. (**B**) *RiboTite(tT/tT)*. (**C**) *cymO/R* and theophylline riboswitch control. *P21*, synthetic P21 promoter. *cmt*, operator sequence of the cumate degradation operon. CymR, CymR regulator. *TheoRS*, theophylline riboswitch. (**D**) *cymO/R* and theophylline ribozyme control. *TheoRZ*, theophylline ribozyme. (**E**) Transcriptional terminator and occluded RBS control. *P_repC_*, promoter regulating *repC*. *TT + oRBS*, transcriptional terminator and occluded RBS. *repC-N*, first 12 nucleotides of *repC.* (**F**) Transcriptional terminator and anti-toxin mRNA antisense RNA control. *P_CON_*, a constitutive promoter. TT, transcriptional terminator. asTOX, antisense RNA against the RBS-start codon region of toxin gene mRNA.

**Figure 2 ijms-21-00705-f002:**
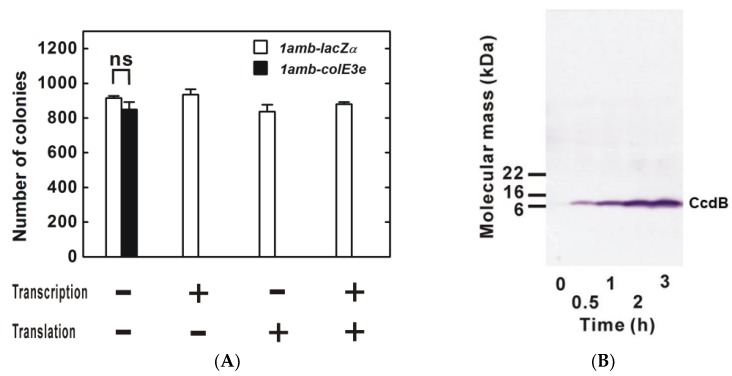
Production of highly-toxic proteins using HYZEL. (**A**) Maintenance of ColE3e expression construct. The expression construct for the *ColE3e* gene, which contains a single amber stop codon insertion, was introduced into *E. coli* BL21-AI with another plasmid that constitutively expresses the specific UaaRS for ZK and its cognate tRNA_CUA_. A V5-LacZ with a gene containing an amber stop codon insertion was used as non-toxic protein control. Leakage expression of ColE3e killed the host bacteria using single repression at the transcriptional or translational level. In contrast, the host bacteria survived in dual transcriptional-translational repression. Data are shown as mean ± s.e.m. of three biological replicates. Statistical analyses were performed using single-factor analysis of variance (ANOVA) with an *α* of 0.05 (ns, not significant). (**B**) Production of the DNA gyrase inhibitor, CcdB. CcdB with a V5 epitope tag added at the N-terminus was produced using HYZEL. An amber stop codon was inserted next to the translation start codon in the V5 epitope tag. The V5-CcdB expression construct driven by T7RNP was cotransformed into BL21-AI with a plasmid that constitutively expressed the specific UaaRS for IY and its cognate tRNA_CUA_. Bacteria carrying these plasmids were cultured overnight in LB medium containing _D_-glucose for catabolite repression against *P_BAD_-araC/O*, which regulates T7RNP gene expression. V5-CcdB production was induced by changing the medium containing IY, _L_-arabinose and IPTG. V5-CcdB production was shown by western blot using an anti-V5 antibody. Time after the medium change is shown below the photo.

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
