# Peer review of "Extremely Low Leakage Expression Systems Using Dual Transcriptional-Translational Control for Toxic Protein Production"

_ijms, 2020, doi:10.3390/ijms21030705_

Round 1

Reviewer 1 Report

I enjoyed very much reading this review. I think it is comprehensive and well written. Once published, I shall suggest the article as reading material for my students of biotechnology.

Reviewer 2 Report

In the review, the author described the use of dual transcriptional-translational control for toxic protein production. However, the aim of the current study is not clear. Additionally, English editing would improve the text, in order to make the goals of the study more clear. 

Author Response

sdfdsf

Reviewer 3 Report

In this Review, the author Yusuke Kato summarizes the principles of dual transcription/translation control system in bacteria. In my opinion the Review can be published, however it should be improved in the following parts.

In my opinion the figure 1. should be eliminated because in the contest of the review the figure 1 is too elementary. 

    2. In the conclusion section the author should give more emphasis at the future perspectives on these system and also including the bottleneck for dual transcriptional/translational control system.

Author Response

dfdfsd

Reviewer 4 Report

Extremely low leakage expression systems using dual transcriptional-translational control for toxic protein

By Yusuke Kato.

The review addresses a very important topic of tight regulation of cloned genes in expression plasmids and minimum expression of toxic proteins. Most widely-used inducible expression systems are regulated at the transcriptional level only. This review discusses the features of dual transcriptional and translational control systems in which extreme suppression of leakage expression can be achieved.

This is a very well written review article with almost no typographical errors. It covers all aspects of the topic and is very informative. The various features of the regulations has been well explained.

Fig 2 line 138 Change “used a” to “used as”

Round 2

Reviewer 2 Report

Despite the efforts of the author, I do not believe that the manuscript has been significantly improved.